# Development of Highly Efficient Resistance to *Beet Curly Top Iran Virus* (*Becurtovirus*) in Sugar Beet (*B. vulgaris*) via CRISPR/Cas9 System

**DOI:** 10.3390/ijms24076515

**Published:** 2023-03-30

**Authors:** Kubilay Yıldırım, Musa Kavas, İlkay Sevgen Küçük, Zafer Seçgin, Çiğdem Gökcek Saraç

**Affiliations:** 1Department of Molecular Biology and Genetics, Faculty of Arts and Sciences, Ondokuz Mayıs University, 55139 Samsun, Turkey; 2Department of Agricultural Biotechnology, Faculty of Agriculture, Ondokuz Mayıs University, 55139 Samsun, Turkey; 3Department of Biomedical Engineering, Faculty of Engineering, Akdeniz University, 07600 Antalya, Turkey

**Keywords:** CRISPR, gRNA/Cas9, sugar beet, BCTIV, *Becurtovirus*

## Abstract

Beet Curly Top Iran Virus (BCTIV, *Becurtovirus*) is a dominant and widespread pathogen responsible for great damage and yield reduction in sugar beet production in the Mediterranean and Middle East. CRISPR-based gene editing is a versatile tool that has been successfully used in plants to improve resistance against many viral pathogens. In this study, the efficiency of gRNA/Cas9 constructs targeting the expressed genes of BCTIV was assessed in sugar beet leaves by their transient expression. Almost all positive control sugar beets revealed systemic infection and severe disease symptoms (90%), with a great biomass reduction (68%) after BCTIV agroinoculation. On the other hand, sugar beets co-agronioculated with BCTIV and gRNA/Cas9 indicated much lower systemic infection (10–55%), disease symptoms and biomass reduction (13–45%). Viral inactivation was also verified by RCA and qPCR assays for gRNA/Cas9 treated sugar beets. PCR-RE digestion and sequencing assays confirmed the gRNA/Cas9-mediated INDEL mutations at the target sites of the BCTIV genome and represented high efficiencies (53–88%), especially for those targeting BCTIV’s movement gene and its overlapping region between capsid and ssDNA regulator genes. A multiplex CRISPR approach was also tested. The most effective four gRNAs targeting all the genes of BCTIV were cloned into a Cas9-containing vector and agroinoculated into virus-infected sugar beet leaves. The results of this multiplex CRISPR system revealed almost complete viral resistance with inhibition of systemic infection and mutant escape. This is the first report of CRSIPR-mediated broad-spectrum resistance against *Becurtovirus* in sugar beet.

## 1. Introduction

Beet curly top disease (BCTD) is a pathogenic viral infection that causes significant yield limitations for sugar beet (*Beta vulgaris*) in arid and semi-arid regions of the world. Historically, *Beet curly top virus* (BCTV, the genus *Curtovirus*) was reported to be the first main causative viral agent for BCTD [1]. It was first recognized in the late nineteenth century in the Western parts of the US and caused a devastating effect on sugar beet production until resistant cultivars became available in the mid-1930s [2].Utilization of the resistant cultivars with insecticide treatments strongly reduced BCTD incidence (Strausbaugh et al., 2017). Nevertheless, infection of the resistant sugar beet cultivars at early stages has been reported to increase the incidence of the disease and cause significant yield limitations in recent years [3]. In the last decades, a divergent curly top virus, *Beet Curly Top Iran Virus* (BCTIV), has also been reported to be a dominant and widespread virus for the sugar beet in the Middle East (e.g., Iran and Turkey) [4,5,6,7,8]. The BCTIV infection caused significant annual damage and yield reduction for sugar beet production in the region. BCTIV has very similar curly top symptoms to curtoviruses in the sugar beet, such as enation, leaf curling, and stunted growth [5]. On the other hand, the genome sequence of BCTIV was reported to be highly divergent from that of curtoviruses (<60% genomic similarity). The virion-sense strand genes are more similar to becurtovirus, while the genes in the complementary-sense orientation encoding Replication protein (REP) from a spliced transcript are similar to mastreviruses (Figure 1) [9]. All this genomic diversity enabled BCTIV to be included in a new genus: *Becurtovirus* under Geminiviridae [9].

BCTD was reported to cause a significant yield reduction in sugar beet and other crops with the increase in global warming in the Middle East and Mediterranean basin [5,8]. For instance, disease incidence was reported to be more frequent in Turkey [10,11] in the last decades and caused more than 50% yield reductions in sugar beet fields during dry seasons. In our previous study, an extensive sample collection was carried out in the sugar beet fields of Turkey. The viral genomes were isolated and sequenced to identify the BCTD causative agents [8]. The results of the study indicated that BCTIV was the main viral agent responsible for BCTD formation in Turkish sugar beet fields. With this study, the existence of BCTIV species was revealed for the first time outside of Iran. Many studies have stressed that BCTIV-dependent diseases would be more aggressive in the future with the increase in global warming and would be one of the main devastating agents for agricultural production in Middle East [6,7,8,10]. Therefore, an effective strategy should be improved and applied for the inhibition of viral spread and BCTD formation in agricultural plants.

CRISPR/Cas (clustered regularly interspaced short palindromic repeats (CRISPR)/CRISPR-associated genes (Cas)) technology is a simple and efficient gene targeting technology that was utilized for plant resistance against many geminiviruses such as BCTV (Curtovirus) [12],, BeYDV (Mastrevirus), [13], TYLCV (Begomovirus) [14], CLCuMuV (Begomovirus) [15], and ChiLCV (Begomovirus) [16],. In these studies, transient or stable expression of gRNA/Cas9 constructs targeting the viral genomes revealed almost complete resistance with no obvious disease symptoms [17,18,19]. The results of all these studies have demonstrated that CRISPR/Cas system offers enormous potential to obtain BCTD-resistant plants either by direct cleavage of the BCTIV genome or by modifying the plants’ genome itself to introduce viral immunity. In the current study, we aimed to design a transient expression analysis for the determination of the most effective gRNA/Cas9 constructs targeting the expressed portion of the BCTIV genome (replication, capsid, movement, and ssDNA regulator genes) and inhibiting its replication and accumulation in sugar beet. Constructs having multiple gRNA/Cas9 cassettes to target all the genes of BCTIV simultaneously were also designed and tested in this transient expression assay to reach the highest viral resistance level in sugar beet.

## 2. Results

In the current study, 8 gRNAs targeting the BCTIV genes and their overlapping parts were designed and transferred into Cas9 containing Agrobacterium plasmid (pHSE401). Each gRNA/Cas9 construct and BCTIV was agroinoculated into the mature sugar beet leaves for their simultaneous transient expression. Before this experiment, the transient expression level of Cas9 was measured at different time intervals (6-12-24-48-120-240-480 h) by qPCR assays. Cas9 expression reached its highest expression level after 24 h of agroinoculation and stayed at this expression level for five days. Most of the agroinoculated leaves were dead after two weeks of agroinoculation. Therefore, gRNA/Cas9 constructs were firstly agroinoculated to the below (close to the petiole) part of two mature leaves (local). One day later, BCTIV containing Agrobacterium was inoculated to the upper parts of the same local leaves. After inoculation, four sugar beet plants per treatment were completely harvested weekly. Agroinoculated mature (local) leaves and newly emerged non-inoculated leaves (systemic) were separated for DNA isolation.

### 2.1. RCA Assay Indicated Lower Translocation of BCTIV from gRNA/Cas9 Treated Local Leaves to Newly Emerged Systemic Leaves

Replication of BCTIV on local leaves and its spread to systemic leaves was firstly measured within these isolated DNAs via RCA assays. As shown in Figure 1, the virus genome successfully circulated and replicated itself in the upper parts of all agroinoculated local leaves. Viral replication, translocation, and accumulation in systemic leaves were also followed for five weeks after agroinoculation. Viral spread from local to systemic leaves was first observed in control plants after the first week of agroinoculation. After running the RCA products on the gel (Figure 2), it was recorded that the viral spread from local to systemic leaves was delayed and decreased in gRNA/Cas9 agroinoculated sugar beets. As can be seen in Figure 2, 18 out of the 20 BCTIV agroinoculated control plants (90%) had viral translocation from local to systemic leaves. This rate significantly decreased to 10–55% with additional gRNA/Cas9 agroinoculation onto the BCTIV-treated sugar beet leaves. Only four out of the 20 agroinoculated plants (20%) with the gRNA/Cas9 constructs targeting V2 revealed RCA bands indicating virus translocation. The inhibition rate of viral translocation for the gRNA/Cas9 constructs targeting V1 and V3 was estimated to be 45% and 30%, respectively. gRNA/Cas9 constructs targeting the overlapping region between functional genes (V1/V2 and V2/V3) indicated the best performance in terms of the inhibition of the virus spread from local to systemic leaves (Figure 2). Viral translocation from local to systemic leaves decreased to 10% and 15%, when the overlapping region between V2/V3 and V1/V2 was targeted with gRNA/Cas9 systems. The replication region of BCTIV was targeted with three gRNA/Cas9 constructs and the RCA assay indicated a reduction to 25%, 35%, and 55% virus translocation for gRNA/Cas9 systems targeting the replication A (C1), replication B (C2), and the overlapping region between C1 and C2, respectively (Figure 2).

### 2.2. qPCR Assay Represented the Lower Accumulation of BCTIV in gRNA/Cas9-Treated Local and Systemic Leaves

The intensity of the RCA products was increased in the systemic leaves of control plants through the end of the experiments, which revealed an increase in viral accumulation over time (Figure 2). On the other hand, the band intensity obtained from the local and systemic leaves of gRNA/Cas9 treated sugar beets was visualized to be much lower compared to only BCTIV agroinoculated ones, indicating the lower accumulation of the virus in CRISPR-treated plants. All the results of the RCA assay revealed the high efficiency of the gRNA/Cas9 systems on the inhibition of BCTIV replication and spread in sugar beet. However, RCA reactions might give false positive results due to the circular nature of mitochondrial and chloroplast DNA, or may not give any result due to the lover amount of viral DNA or experimental errors. Therefore, viral replication, accumulation, and spread of BCTIV in control and gRNA/Cas9 treated sugar beets were also verified with qPCR-based viral quantification. The qPCR assays represented very similar results to RCA assays and verified the inhibition efficiency of the gRNA/Cas9 system on the BCTIV accumulation and spread in sugar beet. The qPCR-based quantification was first carried out on the BCTIV and gRNA/Cas9 agroinoculated local leaves. Viral quantity in the systemic leaves of the control plants was measured to be much higher at the end of five weeks compared to the viral load of the local leaves. This represented the replication and accumulation of the virus in newly emerged leaves and meristematic part of the sugar beet in time. As can be seen in Figure 3A, the relative amount of virus in the gRNA/Cas9 treated local leaves reduced to 5–60% compared to only BCTIV-treated control plants. Plants treated with gRNA/Cas9 constructs targeting the movement gene (V2—25%) and its overlapping region between capsid (V1/V2—25%) and ssDNA regulator (V2/V3—12%) represented the lowest viral accumulation in the local leaves compared to control (100%). Similar to RCA results, no qPCR amplification had been recorded in the systemic leaves of gRNA/Cas9 treated sugar beets in the first week of agroinoculation (Figure 3B), which indicates a delay in viral translocation and disease formation. The viral accumulation in systemic leaves of sugar beets treated with gRNA/Cas9 decreased to 5–60% at the end of the experiment compared to the local leaves of the control plant.

### 2.3. PCR-RE Assay Revealed the High gRNA/Cas9 Efficiency on the Knockout of BCTIV Genome and No Mutant Escape from Local to Systemic Leaves

The existence of restriction enzyme (RE) cleavage sites on the gRNA target site of the viral genome was used for the identification of CRISPR-induced mutation. The RCA products obtained from agroinoculated local leaves of sugar beet were used for PCR amplification and RE digestion. The uncut fragments in the gel electrophoresis indicated the CRISPR-mediated mutation in the target viral site and its NHEJ repair by the host plant cell. PCR-RE assay revealed the presence of mutated fragments for each gRNA/Cas9 system (Appendix A). PCR products were then Sanger sequenced five times in two directions. In this way, 80 sequencing data were obtained from the virus in local leaves to investigate and verify the NHEJ-mediated mutation in the gRNA/Cas9 treated BCTIV genome. All the sequence data revealed the mutation in the target region (after three nucleotides from the PAM sequence) for each gRNA/Cas9 (Appendix A). A total of 31 different mutations were detected for the targeted regions in the BCTIV genome. While the majority of these mutations were deletion (64%), the rates of insertion and base change (A < T, G < C, etc.) were 22% and 14%, respectively (Appendix A). The sequence data were also analyzed with TIDE software to detect the genome editing efficiencies of each gRNA. TIDE can detect overall INDEL frequencies and generate a priori information if the introduced mutations are significant or not. As can be seen in Appendix A, the total efficiency of the gRNAs calculated by TIDE ranged between 52.7% to 88.2%. TIDE analysis detected the mutations in exact position (3 pb after PAM seq) for each sequence data and calculated high gRNA efficiencies for the REP(C1—69.9%, C2—68%) capsid (V1—67%), movement (V2—65.4%) and ssDNA regulator (V3 52.7%) genes of BCTIV. Similar to the RCA results, the highest total efficiency (88.2%) was recorded for the gRNA targeting the overlapping region between movement and ssDNA regulator region (V2/V3). Total efficiencies of gRNAs targeting the other overlapping region between V2/V1 and C1/C2 were 70.7% and 52.9%, respectively. The results of the TIDE corresponded well with RCA assays and indicated the high efficiency of designed gRNA/Cas9 systems in terms of catching the virus and knockout its genome. 

Despite the high inhibitory effects of CRISPR systems on BCTIV replication and spread, all the gRNA/Cas9 treated sugar beets showed virus translocation from local to systemic leaves. This situation could be explained by virus or mutant escape from the gRNA/Cas9 system. Therefore, RCA products obtained from the systemic leaves of gRNA/Cas9 treated sugar beet were used for PCR reaction to amplify the target regions. Each PCR reaction was then treated with a specific restriction enzyme to identify the gRNA/Cas9-mediated mutation. In this way, 47 RCA products obtained from the systemic leaves of sugar beets were analyzed and almost all the results of the PCR-RE assay indicated no mutation in the viral genome. Only three PCR-RE reactions obtained from REP-A/B targeted viral region revealed a slight extra non-cut band in the gel. Sequencing of these bands revealed single base substitutions (A < T, G < C, etc.) which caused a silent mutation in the viral genome. All these results indicated that the efficiency of the gRNA/Cas9 system is highly dependent on its compatibility with the viral genome and the functionality of the targeted gene.

### 2.4. BCTD Symtoms and Biomass Reduction Decreased with gRNA/Cas9 Treatment in Sugar Beets

Development of beet curly top disease (BCTD) was also followed for agroinoculated sugar beets in terms of disease severity and biomass reduction. During disease formation, the leaf margins of BCTIV-positive control sugar beets curled and the veins of the leaves became thicker at the mild disease stage (Figure 4). The rate of BCTIV positive controls plants having mild or no BCTD symptoms were estimated to be 20% while the rate of the same group increased to 75–100% in gRNA/Cas9 treated sugar beets (Table 1). Rolling of the leaf margins downwards and reduction in growth were accepted as moderate disease symptoms in the study (Figure 4). BCTIV sugar beets having severe disease symptoms indicated dwarfed, crinkled and completely rolled leaves, while the underside of leaves was roughened and often produce spike-like outgrowths (red arrow in Figure 4). Among the BCTIV-agroinoculated sugar beets, 14 plants died at the younger stage while survived ones exhibited generally moderate to severe (80%) leaf curling associated with stunted growth (Figure 4).

Almost all the gRNA/Cas9 treated sugar beets survived, and the rate of plants having moderate to severe BCTD symptoms decreased to 0–35% (Table 1). Most of the gRNA/Cas9 treated plants did not revealed any BCTD symptoms. Among the BCTIV positive control plants, only two sugar beets had no BCTD symptoms, while the rate of healthy plants increased to 45–85% in gRNA/cas9 treated counterparts. Among the single gRNA/Cas9 treated plants, the highest number of healthy plants (85%) was recorded for the gRNA/Cas9 construct targeting the overlapping region between movement and ssDNA regulator (V2/V3) (Table 1). The reduction in plant biomass for the BCTIV-inoculated positive control plants was estimated to be 68% compared to mock vector inoculated ones. The same comparison for the gRNA/Cas9 treated sugar beets decreased and ranged between 13% and 45% (Table 1).

### 2.5. Multiplex CRISPR System Had Full Resistance to BCTIV in Sugar Beet

Although gRNA/Cas9 constructs targeting the one viral gene had significant efficiency on the inhibition of viral replication and spread, the BCTIV transfer from local to systemic leaves was recorded for all gRNA/Cas9 treated sugar beets. RCA-, qPCR-, and gRNA/Cas9-mediated targeted mutation analysis on the viral genome revealed that the inhibition of viral replication and spread is highly depended on the complementarity of gRNA/Cas9 to the target region and functionality of the target gene. Therefore, gRNAs having the highest viral inactivation were used to design a multiplex CRISPR vector to obtain full resistance against BCTIV in sugar beet. Four gRNAS targeting the overlapping regions between movement/capsid (V1/V2) and movement/ssDNA regulator (V2/V3) as well as RepA and B were included into a Cas9 containing plasmid and agroinoculated into the BCTIV-treated leaves. In this way, all the genes in the BCTIV genome were targeted with this multiplex CRISPR system. The RCA assay on the systemic leaves of these sugar beets produced a very slight band only in one plant (5%) and revealed almost complete resistance for all the BCTIV agroinoculated plants (1/20). The band intensity in the local leaves of multiplex CRISPR-treated sugar beets was much lower compared to positive control and other gRNA/Cas9-treated sugar beets. qPCR-based viral detection also verified BCTIV transfer from local to systemic leaves in only one plant. As can be seen in Figure 5, the BCTD symptoms such as rolling and yellowing of the leaves in the BCTIV inoculated sugar beets was recorded after the first week of agroinoculation (red arrows). The disease symptoms became more severe at the end of the experiment, and most of the BCTIV inoculated positive plants were dead or had complete leaf curling and stunted growth. On the other hand, 90% of the multiplex CRISPR treated sugar beets did not have any BCTD symptoms and almost no reduction was recorded in biomass compared (11%) to the negative control counterpart. This result indicated that multiple CRISPR systems effectively neutralize the BCTIV and disease development by capturing and knocking out its genomes.

## 3. Discussion

Geminiviruses (*family Geminiviridae*) are the group of viruses that constitute the largest plant pathogens worldwide [20]. They pose a significant threat to global food security by causing heavy losses in food and crops, especially with global warming. It has been reported that Geminivirus-based epidemics have increased their impact and seriously affected sustainable agriculture in the last two decades. New species and members are constantly being added to this *Geminivirus family*, which currently includes 9 species, with new recombination events and changes in the spread mechanisms of viruses [21]. The latest addition to this family is the species known as Beet curly top Iran virus (BCTIV), defined as a new genus under the name *Becurtovirus* [9]. In our previous study, the existence of BCTIV species was revealed for the first time in Turkey. The study also indicated viral spread to the Middle East, which strengthens the possibility of similar disease reports in many regions, including Europe, soon [8].

The CRISPR system is an effective immune defence system for bacteria. After viral infection, bacteria capture small virus DNA and transfer them into their own genomic DNA to create fragments series known as CRISPR arrays. These arrays are used to develop a viral memory to remember and attack the virus or genomically related viruses. During the viral invasion, the bacteria produce complementary RNA segments for the viral genome to recognize and attach viruses’ DNA with Cas9 endonuclease. The enzyme creates a double-strand break in the virus genome and inhibits its replication and accumulation. CRISPR-Cas9 was adapted from this naturally occurring genome editing system to knockout or modify eukaryotic genes and genomes [22].

The CRISPR/Cas system was recently utilized for plant resistance against several viruses [18,23]. Ji et al. [12] tested the effects of CRISPR/Cas-mediated viral resistance on a BCTV strain inoculated into *Arabidopsis* and tobacco plants. In this study, 43 sgRNA–Cas9 constructs were designed to target and cleave the REP, CP gene, and noncoding regions such as Rep-binding sites and intergenic regions (IR) of BCTV. These constructs were then cloned into Agrobacterium vectors and transiently expressed in the leaves of tobacco by agroinoculation. BCTV was lastly agroinfected on to the leaves of the plants. QPCR-based quantification of viral DNA in the leaves indicated that 38 out of the 43 constructs inhibited viral DNA accumulation by over 60%. Some gRNA/Cas9 constructs targeting the Rep, CP, and IR regions reduced the virus accumulation by more than 93%, and no BCTD symptoms were observed in gRNA/Cas9-inoculated plants. As in all geminiviruses, the Rep protein is responsible for initiating the proliferation reaction of the BCTIV genome. In BCTIV, the genes of the Rep A protein are found in complementary sense strand (Figure 1) and divided into two regions, C1 and C2, by an intronic region [24]. In the current study, complementary strands of BCTIV were targeted with three different gRNA/Cas9 constructs. RCA- and qPCR-based quantification results revealed that gRNA/Cas9 systems targeting the C1, C2, and intronic regions (C1/C2) reduced the viral content to 25%, 35%, and 55%, respectively.

Despite this significant success in terms of inhibition of viral replication and spread, gRNAs targeting the complementary sense strand of BCTIV revealed lower efficiency compared to those targeting the virion sense strand genes. In particular, gRNAs targeting the intronic region between C1 and C2 exhibited the lowest performance among all gRNAs in terms of the rate of plants having systemic infection, BCTD symptoms and biomass reduction. Varsani et al. [1] stressed that the intronic region between C1 and C2 must be cut out to produce a functional Rep protein. Therefore, lower gRNA efficiency in this region could be associated to its lower functionality on virus replication. There are three protein-coding genes in the virion sense strand of BCTIV: capsid protein, movement protein, and ssDNA regulator. The capsid protein (V1) coats the virus genome and transfers it from plant to plant by vector [25]. The movement proteins denoted by V2 are known to be functional in systemic viral infection. The final gene on the virion strand, known as ssDNA or V3, is believed to be a regulatory protein and is thought to be particularly important throughout the virus’ replication [6]. All gRNA/Cas9 systems created for the virion-sense component greatly suppressed virus accumulation and spread in sugar beet. In a transient expression assay, BCTIV genes in tobacco revealed that Rep, together with C1, are the main pathogenicity factors which cause typical viral leaf curling symptoms. In addition, V2 caused a mild leaf curling, thickening, and asymmetric leaves, while V1, V3, and C2 had no clear effect on plant phenotype [26]. In another study, Bahari et al. [27] mutated the BCTIV V2 and concluded that the gene plays a crucial role In viral pathogenicity and systemic movement by suppressing host post-transcriptional gene silencing (PTGS) mechanisms. Similarly, Ebrahimi et al. [28] found that V2 and Rep were able to suppress specific PTGS mechanisms, while BCTIV-V2 could suppress S-PTGS more efficiently than BCTIV-Rep. Luna et al. [29] revealed that V2 from BCTV is able to induce systemic symptoms and necrosis associated with a hypersensitive response, and that this pathogenicity activity is not dependent on its ability to suppress PTGS. In the current study, the gRNAs targeting BCTIV-V2 and its overlapping region between V1 and V3 represented the best performance in terms of systemic infection, BCTD development, and biomass reduction. Therefore, gRNA/Cas9 success seemed to be highly dependent on the functionality of the target gene on viral infection.

In the current study, PCR-RE assay demonstrated NHEJ-mediated mutations in the virus genome for all gRNA/Cas9s. Sequencing data and its analysis with TIDE represented high gRNA efficiencies with INDEL mutations on viral target sites. Despite the high gRNA efficiencies (52.7–88.2%), severe disease formation and viral transfer from local to systemic leaves was also demonstrated in gRNA/Cas9 agroinoculated sugar beets. There could be two reasons for this situation. The first is the replication and infection of mutant viruses. Previous studies have claimed that mutant viruses could continue to spread and accumulate in the plants. In fact, there are doubts that these mutant escapes could result in the creation of more aggressive virus strains [15,30,31,32,33]. Some other studies predicted that gRNA/Cas9 systems that specifically target non-gene coding regions can destroy viruses without mutant escape [14]. In the current study, viral genomes isolated from systemic leaves indicated no INDEL mutations. Only silent base substitution was recorded in three samples after Sanger sequencing in the viral genome. This results indicated that mutant escape and its replication is not the case for gRNA/Cas9-treated BCTIV. The second reason is lower transient expression of gRNA/Cas9 complex in sugar beet leaves. RT-qPCR assays indicated that Cas9 expression started to decrease after five days of agroinoculation. Therefore, the amount of logarithmically self-replicating virus may break gRNA/Cas9 efficiency and enable viral transfer to the systemic leaves. Similar results have also been reported in previous studies [12,13,14]. However, the systemic infection rate of the gRNA/Cas9-treated virus among sugar beet leaves ranged from 10% to 55% in the current study. This variation could also depend on the knockout capacity of gRNAs on the viral genome and their harmony with tracRNA, as well as their agroinoculation efficiencies and the functionality of the target genes on viral replication and spread.

In the current study, the four most effective gRNAs targeting C1, C2, and two overlapping regions between V1/V2 and V2/V3 were merged in a single vector. Transient expression of multiplex CRISPR in sugar beet revealed almost complete viral resistance against BCTIV. Mutant or viral escape from local to systemic leaves was not recorded in multiplex CRISPR-treated sugar beets. There were no BCTD symptoms and biomass reduction in CRISPR-treated plants. The high efficiency of the multiplex CRISPR systems against replication and accumulation of BCTIV corresponded well with the results of previous studies. Roy et al. [16], for instance, first tested the effects of single gRNAs on the inhibition of chili leaf curl virus (ChiLCV) replication, and then investigated the elimination of the ChiLCV genome with double combination of these gRNAs. The results of this study indicated that multiplex gRNA combinations targeting the overlapping region in the complementary and virion sense strands of the ChiLCV genome (*Begomoviruses*) conferred successful resistance. Roy et al. [16] did not record any viral or mutant escape from CRISPR-treated plant leaves. Yin et al. [15] developed transgenic tobacco plants expressing Cas9 and sgRNAs simultaneously targeting two different sequences in the genome of cotton leaf curl Multan virus (CLCuMuV). The experimental results revealed the plants were completely resistant to CLCuMuV [15]. Another study represented gRNA/Cas9-mediated resistance against wheat dwarf virus (WDV) in barley. The sgRNA-Cas9 construct targeting the multiple sites within conserved regions of two WDV strains effectively eliminated the viral replication and spread to the plant [30]. In another study, strong virus resistance was also achieved against cauliflower mosaic virus (CaMV), with the expression of multiple sgRNAs targeting the coat protein region in transgenic Arabidopsis plants [31].

## 4. Materials and Methods

### 4.1. Plant Material and Infectious Virus Construct

In the current study, an infectious Turkish BCTIV isolate (Accession number MT459431) previously cloned into an Agrobacterium plasmid [8] was used to create BCTD in the plants. Briefly, infected sugar beets having BCTD symptoms were collected from Ankara, and their total DNA was isolated according to Yildirim et al., [34]. The whole genome of the BCTIV Turkish isolate was amplified by using the rolling-circle amplification (RCA) kit (TempliPhi, GE Healthcare, Chicago, IL, USA) from this total DNA as described by Shepherd et al. [35]. The RCA product was digested with HindIII to obtain one unit virus genome. The viral genome was transferred into pUC57 for whole genome sequencing. The sequencing results were aligned with other BCTIV genomes in the database by the blast and alignment tools. Head-to-tail dimer (two consecutive genomes) of BCTIV Turkish isolate was then inserted into the Agrobacterium pBin19 vector by using the In-Fusion Cloning Kit (Takara Inc. San Jose, CA, USA) as described in the manufacture protocol. The final recombinant pBin19 construct was transferred into *Agrobacterium tumefaciens* strain GV3101 by electroporation. When the pBin19-BCTIV construct was agroinoculated onto the mature sugar beet (*Serenada cultivar*) leaves, typical curly top diseases symptoms such as leaf curl, enations, vein roughness, and thickening with leaf dwarfing emerged on 90% of inoculated plants [8].

### 4.2. Designing the gRNAs Targeting the BCTIV Genome

The BCTIV genome contains three overlapping protein-coding genes (V1, V2, and V3) in the virion-sense strand and two genes (C1 and C2) in complementary-sense orientation (Figure 1). The virion-sense strand genes (V1, V2, and V3) encode the coat protein (CP), movement protein (MP), and ss/ds DNA regulating protein, respectively. On the other hand, C1 and C2 produce Replication protein (REP) from a spliced transcript [1]. In the current study, these expressed five genes and their overlapping parts in the BCTIV genome were targeted with gRNA/Cas9 constructs. Therefore, the full genome sequence of the BCTIV Turkish isolate was first aligned with other BCTIV genomes found in databases to identify common sequences on the genes. Then the whole genome of the BCTIV Turkish isolate was uploaded into CRISPRdirect (https://crispr.dbcls.jp/ (accessed on 12 October 2020)) software to identify gRNAs (23 bp) having -NGG PAM sequence. Selected gRNAs were then loaded into RNAfold webserver to test their 3D harmony with tracRNA and Cas9 enzyme as described in Aksoy et al., [17]. Suitable gRNAs having appropriate GC content (<40%) and RE enzyme recognition site in Cas9 cutting region were selected and uploaded into MEGA software to select the common gRNAs for all BCTIV genomes found in databases. The selected gRNAs were loaded to Cas-OFFinder (http://www.rgenome.net/ (accessed on 24 October 2020)) webtool to screen potential off-targets on the *Beta vulgaris* reference genome. After passing all these elimination processes, eight gRNAs that can knock out the BCTIV genome were decided in the study (Figure 1 and Appendix A). The targeted five expressed regions and the number of gRNAs designed for these regions are as follows: one gRNA for coat protein (CP—referred to as V1), one gRNA movement protein (MP—referred to as V2), one gRNA for ss/ds DNA regulating protein (ssDNA—referred to as V3), and two gRNAs for Replication proteins (REP—referred to as C1 and C2). In addition, each overlapping region between V1–V2, V2–V3, and C1–C2 genes was targeted with one gRNA (Figure 1).

After selection of the gRNAs, PAM sequences were excluded and adaptors were added to the 5′ (ATTG–) and 3′ (–CAAA) end of the gRNAs. Overlapping forward and reverse primers were synthesized and then dephosphorylated with T4 Polynucleotide Kinase enzyme (New England Biolabs, Ipswich, MA, USA) to have double-stranded gRNAs. Each gRNA was ligated into pHSE401 CRISPR vector (provided from ADDGENE) and transferred to *E. coli* as described in Secgin et al., [36]. The transformed single colonies were selected by growing them on a solid selective LB media containing 50 mg/L kanamycin, 50 mg/L gentamicin, and 10 mg/L rifampicin under 28°C for two days. After colony PCR and sanger sequencing, recombinant plasmids were isolated from *E. coli* using Bacterial Plasmid Isolasyon kit (FavorGen, Wien, Austria) and transformed into *Agrobacterium tumefaciens* (GV3101) by electroporation according to the protocol described in Seçgin et al. [37]. After the growth of Agrobacterium cells on the selection media, single colonies were incubated in liquid selective media until the bacterial density reached OD_600_ 0.8. The bacterial solution was centrifuged, and the cell precipitate was stocked in 80% glycerol at −80 °C.

### 4.3. Development of Multiplexed gRNA-Cas9 Modules for Efficient Knockout of BCTIV

In the current study, the BCTIV genome was also targeted with a multiplexed gRNA/Cas9 construct to enhance inhibition of the viral replication and spread into the plant. Therefore, four gRNAs targeting the REP protein genes (C1 and C2) and overlapping regions between MP/CP (V1/V2) and MP/ssDNA (v2/v3) were inserted into a Cas9 containing Agrobacterium plasmid with a golden gate cloning as described in Xing et al., [38]. Firstly, a multiplex PCR reaction was carried out with destination vectors, and the adaptor added forward-reverse gRNA primers with a protocol described in Appendix A. The amplified PCR products were visualized in an agarose gel, and each product was extracted from the gel with GeneJET Gel Extraction Kit (Thermo Fisher Scientific, Waltham, MA, USA). The quality and quantity of the PCR products were measured with a nano spectrophotometer. The golden gate cloning reaction was then carried out with four gRNA-containing PCR products as described in Appendix A. In this reaction, purified PCR product (100–200 ng) was incubated with Cas9 containing plasmid (pHSE401), T4 DNA ligase, BSA, and BsaI restriction enzyme (Thermo Fisher Scientific, Waltham, MA, USA). The recombinant vector was then inserted into competent *E. coli dH5α* with electroporation for PCR and sequence-based confirmation of multiplex gRNA transformation. Verified multiple gRNA containing plasmids were then isolated from *E. coli* and inserted into Agrobacterium GV3101 by electroporation. Agrobacterium was grown in the selection media, and single colonies were verified with colony PCR and then stocked in 80% glycerol as described in the previous section. In this way, a multiplexed gRNA/Cas9 module targeting all the genic parts of the BCTIV was designed and constructed for efficient knockout of the virus genome. 

### 4.4. Expression Analysis of Cas9 by Quantitative Reverse Transcriptase PCR (qRT-PCR)

In the current study, gRNA/Cas9 construct was agroinoculated onto the sugar beet leaves for transient expression. BCTIV was then applied to the leaves to test the inhibitory effects of the CRISPR system on virus replication and spread. Before this experiment, a time-depended transient expression level of Cas9 was estimated to select the best time for BCTIV treatment. Therefore, Cas9 containing Agrobacterium was infiltrated onto the mature leaves of sugar beet, and sampling was carried out in 6-12-24-48-120-240-480 h intervals. Total RNA was isolated by using as described in Yıldırım et al., [39]. The expression level of Cas9 was measured in Agilent qPCR System (Mx3000P, Santa Clara, CA, USA) by using Cas9-specific primers and SsoAdvanced Universal SYBR Green Supermix (Bio-Rad, Hercules, CA, USA) as described in Xing et al., [38]. The sugar beet actin gene was used as an internal control, and relative expression was calculated by using the 2^−ΔΔCT^ method [40]. Each expression was measured in three replicates for all the samples.

### 4.5. Agroinoculation of gRNA/Cas9 Constructs and BCTIV on the Sugar Beet for Transient Expression Assay

The inhibitory effect of the CRISPR system on BCTIV replication and systemic infection was tested by subsequent co-agroinoculation of the sugar beet leaves with one gRNA-Cas9 construct and BCTIV infectious construct as described in Ji et al. [12]. Bacterial cultures having gRNA/Cas9 constructs were prepared from stock solution by growing them overnight in the same liquid selective media. The bacterial cells were centrifuged, and the precipitate was re-suspended in an infiltration buffer (10 mM MES, 10 mM MgCl2, and 150 μM acetosyringone) to reach a final bacterial density OD600~0.5. The culture was kept at room temperature for three hours and used to infiltrate two mature leaves of 2-week-old sugar beet plants using a needleless syringe. Due to the double-stranded nature of the plasmid DNA, gRNA/Cas9 complex could cut the pBin–BCTIV plasmid before its replication and spread from cell to cell. To exclude this possibility, the gRNA/Cas9 containing GV3101 (OD600 = 0.5) was first injected with 0.5 mL of the solution onto the leaf lamina adjacent to the petiole, while the BCTIV infectious construct was infiltrated (1 mL at OD600 = 0.5) to the apex part of the same leaves after one day (24 h). In this way, BCTIV generated by the pBin-BCTIV construct in the top part of the leaf should be able to replicate and spread through the bottom. Therefore, the efficiency of the gRNA/Cas9 constructs was measured by actively replicating the virus rather than on incoming plasmids. In this experiment, a total of 40 sugar beet plants (two plants per pot) were used to evaluate the efficiency of each gRNA/Cas9 construct. Agroinoculation of the BCTIV construct with an empty gRNA-Cas9 vector (no gRNA) backbone to the 40 sugar beets served as a positive control, while agrobacterium inoculation without any vector backbone to the plants served as mock negative controls. Four plants were completely harvested weekly and agroinoculated mature leaves (local) and non-inoculated newly emerged leaves (systemic) were separated and stored at −80 °C until DNA isolation. Plant sampling was carried out for five weeks, and the severity of the BCTD on gRNA/Cas9 treated sugar beets was evaluated for the next four weeks on the remaining 20 plants according to symptoms described in Montazeri et al. [41]. In this evaluation system, BCTIV infection symptoms were scaled according to the curling of leaves and stunting of the sugar beets. In this categorization, 1—healthy plants with no visible symptoms or mild symptoms indicating vein thickening, puckering, and folding of leaves from the margin; 2—moderate symptoms exhibited such as cupping, rolling, and size reduction in sugar beet leaves; 3—severe symptoms represented, such as stunting of sugar beets with complete leaf curling and spike-like structure formation. In addition to BCTD symptoms, the biomass of the gRNA/Cas9 agroinoculated plants was also measured throughout the experiment, and the results were compared with their positive (only BCTIV agroinoculated) control counterparts.

### 4.6. Testing the Efficiency of gRNA/Cas9 on BCTIV Replication and Spread in Sugar Beet

Test of gRNA/Cas9 efficiency on BCTIV spread in sugar beet via RCA assay

Due to the systemic infection of BCTIV from leaf to leaf, viral accumulation and spread ratio were measured on agroinoculated sugar beet leaves. In this experiment, gRNA/Cas9+BCTIV agroinoculated mature leaves (local) of the sugar beet and newly-emerged non-agroinoculated (systemic) leaves of sugar beets were used to estimate the systemic infection ratio compared to only BCTIV agroinoculated control plants. For this purpose, isolated DNAs from the local and systemic leaves were utilized for rolling-circle amplification (RCA) assay (TempliPhi, GE Healthcare, Chicago, US) as described by Shepherd et al. [35]. In this assay, circular viral genomes were firstly replicated with phi29DNA polymerase to produce high molecular weight RCA amplicons. Each replicon was then cut with Hind-III restriction endonuclease and run on the gel electrophoresis to estimate viral existence and accumulation in the local and systemic leaves (Figure 2).

qPCR-based detection of the gRNA/Cas9 efficiency on BCTIV replication and accumulation

Quantitative PCR (qPCR) was used to estimate the replication and accumulation of BCTIV in gRNA/Cas9 agroinoculated mature and systemic leaves compared to their viral-positive control plants. Genomic DNA was extracted from the agroinoculated mature and newly emerged leaves from four weekly-harvested sugar beets using a DNeasy Plant DNA extraction kit according to manufacturer protocol (Quigen, Hilden, Germany). An equal amount of each DNA sample was then mixed and used for qPCR-based quantification of viral accumulation. Before viral quantification in the sugar beet leaves, BCTIV titers were assayed by qPCR as previously described by Gadiou et al. [42]. The absolute quantification of viral DNA copies was performed by using an Agilent qPCR System (Mx3000P, Santa Clara, CA, USA). The PCR Master Mix comprised six μL of SYBR Green Supermix (Bio-Rad, Hercules, CA, USA) and 0.6 μL of the primer pair mix in a final concentration of 10 μM. The thermal cycling protocol was started with an initial denaturation at 95 °C for 10 min. Then 40 cycles were carried out at 95 °C for 15 s, 60 °C for 1 min, 95 °C for 15 s, and 60 °C for 15 s. The fluorescence was measured via a 60–97 °C melting curve. The BCTIV titer in each sample was calculated with a standard curve prepared by the pBin19-BCTIV construct. The quantification of viral DNA copies was done according to the formula described in Jarosova et al. [43]. qPCR assay was performed in three replicates, and the sugar beet actin gene was used as a reference control during calculations. In the study, viral accumulation in only BCTIV agroinoculated local leaves was used as a positive control to estimate the relative accumulation of the virus in gRNA/Cas9 inoculated local and systemic leaves.

Mutation detection on BCTIV genome by loss of restriction site assay and Sanger sequencing

In the current study, a loss of restriction site assay was utilized to determine the targeted genome modification created by the activity of the gRNA/Cas9 complex and double-strand break repair of the host through the non-homologous end joining (NHEJ) pathway. As represented in Appendix A, each gRNA was designed to have a specific restriction enzyme (RE) recognition site. RCA products obtained from local and systemic leaves of the sugar beets were used for PCR-based amplification of genomic fragments carrying the gRNA/Cas9 target site. Therefore, RCA products obtained from local and systemic leaves of four plants were mixed and used for PCR amplification with specific primer pairs to obtain fragments encompassing the gRNA/Cas9 targeted genome site (Appendix A). In a 20-μL reaction, 300 ng PCR products were subjected to gRNA-specific restriction enzyme digestion according to manufacturer protocols (New England BioLabs). The digested PCR products were run on agarose gel, and the bands of fragments were captured with a gel image system to detect the digested and non-digested bands using ImageJ (Appendix A). The PCR fragments were Sanger sequenced five times in two directions to detect the mutations on the BCTIV genome. The sequence results were analysed through the Tracking of Indels by Decomposition (TIDE) program (https://tide.nki.nl/ (accessed on 20 October 2022)) as described previously in Brinkman et al. [44]. TIDE analysis provides a rapid and reliable assessment of genome editing efficiencies. It quantifies the rates of NHEJ-mediated repair in an edited sample by decomposing the sequence trace data. It identifies the predominant types of insertions and deletions (INDELs) in the DNA. Briefly, the virus sequences from gRNA-Cas9 treated and untreated plants were compared to detect a mutation in the flaking region of the gRNA target site. All analyses were performed with a default setting.

## 5. Conclusions

We provide here the first proof of concept of gRNA-Cas9 modules to inhibit becurtovirus infection through a transient expression assay in sugar beet. The results of the experiment indicated that the gRNA/Cas9s targeting the movement gene and its overlapping region between capsid and ssDNA regulator had high efficiency on inactivation of BCTIV and disease formation. We also designed a multiplexed gRNA/Cas9 module targeting all the functional genes (REP, CP, MP, ssDNA reg.) of BCTIV. Transient expression of this multiplexed CRISPR system in sugar beet leaves enabled almost complete viral resistance with loss of BCTD symptoms and mutant escape. Thus, for the first time, a strategy that provides full resistance to the *Becurtovirus* genome was developed using the CRISPR system directly on sugar beet. The gRNAs were designed according to all BCTIV genomes found in the literature, and they can be used for all becurtovirus strains. Stable expression of these multiplex CRISPR systems on sugar beet or other agricultural plants will strongly enhance resistance against becurtoviruses.

## Figures and Tables

**Figure 1 ijms-24-06515-f001:**
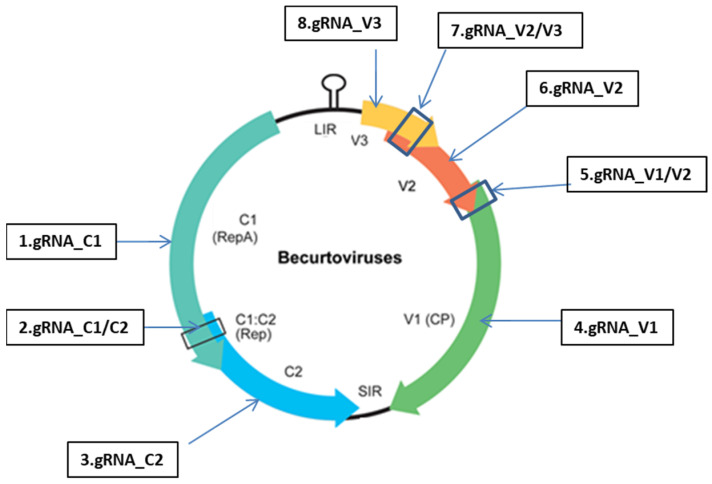
gRNAs targeting the genic and intergenic region of the BCTIV genome. Genic regions encoding coat protein (CP—referred to as V1), movement protein (MP—referred to as V1), ss/ds DNA regulating protein (ssDNA—referred to as V3), and Replication protein (REPA and B—referred to as C1 and C2) were targeted via CRISPR/Cas9 system with the designed gRNAs. Short intragenic region (referred to as SIR) and long intragenic region (referred to as LIR) were also represented in the figure. Detailed information about the gRNA is provided in Appendix A.

**Figure 2 ijms-24-06515-f002:**
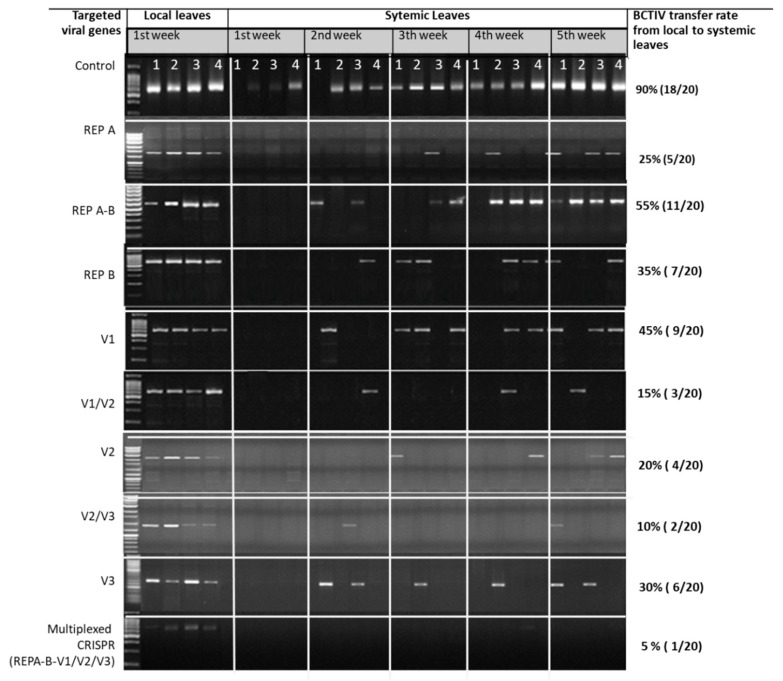
The results of RCA assay in local and systemic leaves of sugar beet co-agroinoculated with BCTIV and gRNA/Cas9. The rate of RCA fragments revealed by the RCA assay was used as an indicator for BCTIV systemic infection from local to systemic leaves. The PCR bands on the gel electrophoresis indicate the whole-length genome of BCTIV (~2840 nt) produced by the RCA reaction in local and systemic leaves of sugar beet. Twenty sugar beets were co-agroinoculated with BCTIV and CRISPR constructs and viral movement from local to systemic leaves were fallowed for five weeks by RCA assay. For each week four plants were harvested for DNA isolation and biomass measurements. The viral transfer was measured as the ratio of RCA products obtained from systemic leaves of sugar beets.

**Figure 3 ijms-24-06515-f003:**
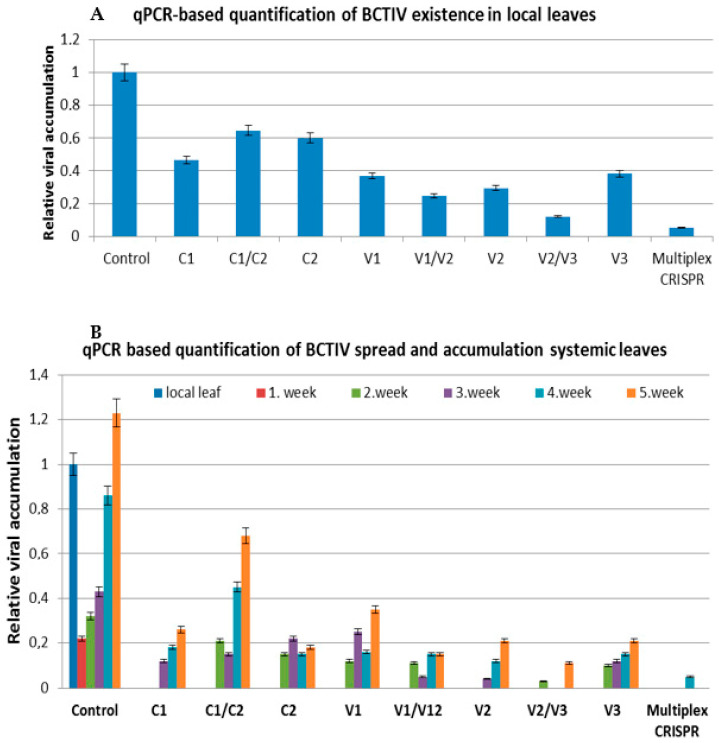
qPCR-based quantification of BCTIV content in gRNA/Cas9 and BCTIV co-agroinoculated local (**A**) and systemic leaves (**B**) in sugar beet. The viral accumulation was quantified according to the viral content of the local leaves in positive control sugar beets collected after the first week of agroinoculation.

**Figure 4 ijms-24-06515-f004:**
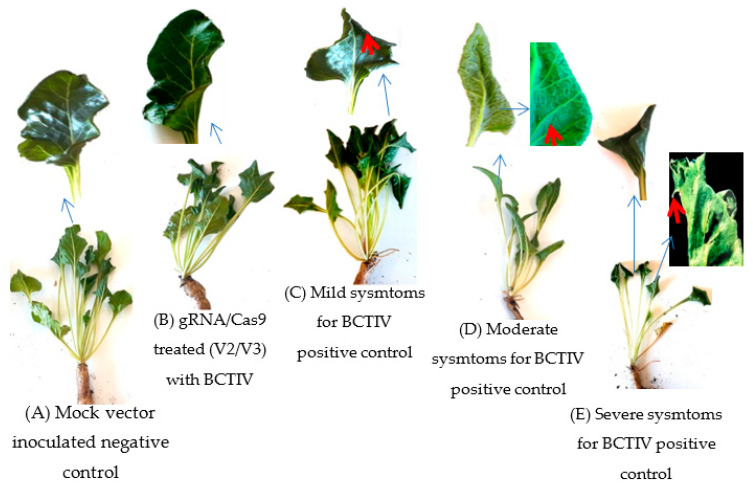
Beet curly top disease symptoms on sugar beets due to the agroinoculation of BCTIV. Mock vector and gRNA/Cas9 inoculated plants represented no disease symptoms (**A**,**B**). On the other hand, BCTIV agroinoculated positive control plants revealed mild leaf curling and thicker vein formation on the leaves (**C**), while most of them indicated moderate (**D**) or severe (**E**) disease symptoms such as dwarfed, crinkled, and completely rolled leaves with spike-like outgrowths (Red arrows).

**Figure 5 ijms-24-06515-f005:**
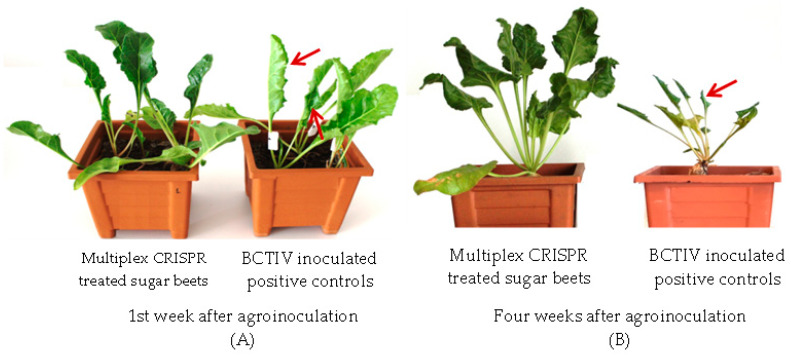
BCTD formation and disease symptoms (red arrow) in multiplex CRISPR treated plants compared to only BCTIV inoculated control plants after first week of agroinoculation (**A**) and at the end of experiment (**B**).

**Table 1 ijms-24-06515-t001:** Biomass and beet curly top disease (BCTD) symptoms were recorded for 40 sugar beets co-agroinoculated with gRNA/Cas9 and BCTIV or only BCTIV and mock vector. The rate of disease formation and biomass reduction was compared with the mock vector-agroinoculated healthy sugar beets.

	Healthy Plants without Symptoms (%)	Plants with Mild Symptoms (%)	Plants with Moderate and Severe Symptoms (%)	Average Weight of Leaves(g)	Average Weight of Roots(g)	Rate of Reduction in Biomass (%)
Mock vector agroinoculated controls	100	0	0	124 ± 12	365 ± 24	-
BCTIV agroinoculated positive controls	5	15	80	36 ± 4	123 ± 12	68
C1-REPA	60	15	25	90 ± 5	256 ± 21	30
C1/C2-REPA/REPB overlapping region	45	30	35	76 ± 11	196 ± 9	45
C2- REPB	75	10	15	96 ± 6	266 ± 14	26
V1- Capsid protein (CP)	65	15	20	88 ± 3	286 ± 24	24
V1/V2-CP/MP overlapping region	85	15	0	106 ± 4	316 ± 16	14
V2- Movement protein (MP)	65	10	25	74 ± 5	275 ± 13	29
V2/V3-MP/ssDNA Reg. overlapping region	70	10	15	108 ± 12	321 ± 18	13
V3 ssDNA regulator gene	75	10	15	84 ± 2	295 ± 9	23
Multiplex CRISPR (C1/C2/V1-V2/V2-V3)	90	10	0	115 ± 9	322 ± 14	11

## Data Availability

Not applicable.

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
