# Peer review of "Development of Highly Efficient Resistance to *Beet Curly Top Iran Virus* (*Becurtovirus*) in Sugar Beet (*B. vulgaris*) via CRISPR/Cas9 System"

_ijms, 2023, doi:10.3390/ijms24076515_

Round 1

Reviewer 1 Report

The manuscript reports the development of gene-editing CRIPSR-mediated resistance to Beet curly top Iran virus (BCITV), an emerging virus of significant importance to sugar beet, using transient delivery of selected CRISPR/Cas9 constructs targeting different parts of the BCTIV genome.

The authors have experimental evidence that CRIPSR/Cas9 constructs used either individually or in multiplex  do inhibit BCTIV accumulation in sugar beet both in locally agroinfiltrated leaves and in systemic leaves. No obvious BCTIV disease symptoms on multiplex CRIPSR/cas9 agroinfiltrated plants were observed while untreated plants displayed severe stunting BCTIV disease symptoms.

The data presented have sufficient novelty and illustrate the potential of gene editing in controlling BCTIV disease in sugar beet. The manuscript would gain in scientific soundness and impact if the authors reported qPCR data from technical replicates from independent experiments,  in order to assess the variability of the CRIPSR constructs in this system. Perhaps by focusing on the most promising constructs giving the strongest BCTIV inhibition of replication and diseases symptoms such as V2/V3 and Multiplex CRISPR constructs (figure 3). The inclusion of a non-infected and untreated (ie non-infiltrated by CRISPR/cas9 constructs) and infiltrated CRISPR/Cas9 constructs with no BCTIV target RNA sugar beet controls would be necessary to conclude on any impact of agroinfiltration in plant development.

While it can be argued that the manuscript would have a stronger impact if the effect of CRISPR/cas9-mediated resistance to BCTIV in stable transgenic sugar beet, I do feel that a revised version of the manuscript could be accepted for publication once these comments have been adequately addressed. 

Author Response

Point 1: The data presented have sufficient novelty and illustrate the potential of gene editing in controlling BCTIV disease in sugar beet. The manuscript would gain in scientific soundness and impact if the authors reported qPCR data from technical replicates from independent experiments,  in order to assess the variability of the CRIPSR constructs in this system. Perhaps by focusing on the most promising constructs giving the strongest BCTIV inhibition of replication and diseases symptoms such as V2/V3 and Multiplex CRISPR constructs (figure 3). The inclusion of a non-infected and untreated (ie non-infiltrated by CRISPR/cas9 constructs) and infiltrated CRISPR/Cas9 constructs with no BCTIV target RNA sugar beet controls would be necessary to conclude on any impact of agroinfiltration in plant development.

While it can be argued that the manuscript would have a stronger impact if the effect of CRISPR/cas9-mediated resistance to BCTIV in stable transgenic sugar beet, I do feel that a revised version of the manuscript could be accepted for publication once these comments have been adequately addressed.

Response 1: Many thanks to the referee’s positive comments and suggestions. This is the first report indicating the inhibitory effects of the CRISPR system on BCTIV infection in sugar beet. We have already started to work on the stable transfer of multiplex CRISPR and the most effective gRNA/Cas9 constructs on sugar beet. We added a paragraph about our further study on the stable transformation of CRISPR systems in sugar beet to the end of the conclusion. During our future works, we will test the effects of stable expression of CRISPR constructs on BCTIV infection and its impact on plant development, off-target formation, yield, and other agricultural traits on the transgenic sugar beets. In this preliminary study, we aimed to identify the most effective gRNA/Cas9 constructs on BCTIV infection compared to non-CRISPR-treated but BCTIV agroinoculated plants. Therefore, our qPCR results were relatively calculated for each gRNA/Cas9 construct and given in the graph. We insist on giving the relative expression of qPCR results for each gRNA/Cas9 construct to also exhibit the function of genes on BCTIV  infection.

Other referees also commented on the qPCR graph to change the shape of the illustration and give a bar graph. We changed the shape of qPCR graph and made it clear. We hope it is now in a more acceptable format. We will consider your suggestions for our stable transformation results.

Reviewer 2 Report

This study assessed the efficiency of CRISPR-based gene editing in improving sugar beet resistance against Beet Curly Top Iran Virus (BCTIV-Becurtovirus), which is a dominant pathogen causing damage and yield reduction in sugar beet production in the Mediterranean and Middle East. The researchers tested gRNA/Cas9 constructs targeting expressed genes of BCTIV in sugar beet leaves and found that co-agronioculated sugar beets had much lower systemic infection, disease symptoms, and biomass reduction than positive control sugar beets. A multiplex CRISPR approach was also tested, with four gRNAs targeting all the genes of BCTIV cloned into a Cas9-containing vector and agroinoculated into virus-infected sugar beet leaves, which revealed almost complete viral resistance with inhibition of systemic infection and mutant escape. The authors should consider improve the data representations and the figures in the manuscript.

Comments:

1.     In Figure 2, there appear to be overlapping numbers on the left side (such as 330, 331…), and the PCR gel pictures seem to have varying levels of exposure, with some being over-exposed (10% and 20%). It would be helpful for the authors to provide information on the sizes of the target bands, and to add a size marker on the left side. Additionally, the authors should improve the figure legend to make it more clear for the description of each lane and each panel.

2.     In Figure 3, the y-axis appears to be overlapped with some random numbers. It may be beneficial for the authors to convert this figure into a regular bar graph, as the 3-D cone plot is not necessary for representing this data.

Author Response

Point 1: The authors should consider improve the data representations and the figures in the manuscript.

  1. In Figure 2, there appear to be overlapping numbers on the left side (such as 330, 331…), and the PCR gel pictures seem to have varying levels of exposure, with some being over-exposed (10% and 20%). It would be helpful for the authors to provide information on the sizes of the target bands, and to add a size marker on the left side. Additionally, the authors should improve the figure legend to make it more clear for the description of each lane and each panel.

  1. In Figure 3, the y-axis appears to be overlapped with some random numbers. It may be beneficial for the authors to convert this figure into a regular bar graph, as the 3-D cone plot is not necessary for representing this data.

Response 2: Many thanks to the referee's positive comments and suggestions. We checked all the illustrations in the manuscript and realized that the numbers on the left of the figures are formed during the conversion of our word document to PDF in the submission process. If you look at the word document you will see that there are no numbers on the figures.

You are correct in this criticism that the PCR gel pictures have varying levels of UV exposure.  As you may guess there were many PCR and qPCR amplifications and gel electrophoresis during the experiments. Different researchers may sometime prefer different intensities to obtain the best visualization. However, it is not possible to realize this exposure difference before bringing them together into the same figure. We added new describtion to the figure legend. In fact, all bands on the gels indicate the whole-length genome of BCTIV (~2840 nt) produced by the RCA reaction in local and systemic leaves of sugar beet. Therefore there is no different panel and lane in the figure.

We changed the shape of qPCR graph into bars. Again there is no random numbers on the figure. These line numbers are produced during the converstion of word to PDF.

Reviewer 3 Report

BCTD could cause a significant yield reduction in sugar beet and other crops in the Middle East and Mediterranean basin. The current manuscript descripted the gRNA-Cas9 and multiplexed-gRNA/Cas9 modules to inhibit BCTIV infection through a transient expression assay in sugar beet. It’s a very good story with well design and writing. I advise an acceptance subjected to some minor revisions.

1. Line 54 delete and?

2. Line 97 (TempliPhi, GE Healthcare, USA), please check all the reagents and instruments if they are in the right and uniform format.

3. Line 108 %90 should be 90%, please check everywhere else.

4. Line209 be able to 

Author Response

BCTD could cause a significant yield reduction in sugar beet and other crops in the Middle East and Mediterranean basin. The current manuscript descripted the gRNA-Cas9 and multiplexed-gRNA/Cas9 modules to inhibit BCTIV infection through a transient expression assay in sugar beet. It’s a very good story with well design and writing. I advise an acceptance subjected to some minor revisions.

Point 1:

  1. Line 54 delete and?
  2. Line 97 (TempliPhi, GE Healthcare, USA), please check all the reagents and instruments if they are in the right and uniform format.
  3. Line 108 %90 should be 90%, please check everywhere else.
  4. Line209 be able to

Response 3: Many thanks to the referee's positive comments and suggestions. All the mistakes were corrected according to refrees’ comments. The manucript was read and revised completely.

Reviewer 4 Report

Overall, this is the great article. I very much enjoyed to read it. In my personal opinion it can be published as it is without any further improvement.

Author Response

Point 1: Overall, this is the great article. I very much enjoyed to read it. In my personal opinion it can be published as it is without any further improvement.

Response 4: Many thanks to the referee's positive comments.

Round 2

Reviewer 1 Report

The overall content of the manuscripts and added information following reviewer's instructions are suitable and justify publication in IJMS once minor language editing have been performed.

Reviewer 2 Report

The authors addressed my concerns. The manuscript can be accepted.